



# Standardized datasets of Brazilian reef diversity in space and time

André L. Luza[1], Cesar A.M.M. Cordeiro[2], Anaide Aued[3], Diego R. Barneche[4,5], Jessica Bleuel[6], Carlos E.L. Ferreira[7], Sergio R. Floeter[3], Ronaldo B. Francini-Filho[8], Jean-Christophe Joyeux[9], Guilherme O. Longo[6], Thiago C. Mendes[7], Hudson T. Pinheiro[8], Juan P. Quimbayo[8,10], Natália C. Roos[6], Barbara Segal[3], Mariana G. Bender[1]

[1]Marine Macroecology and Conservation Lab. Departamento de Ecologia e Evolução, Universidade Federal de Santa Maria, Santa Maria, Rio Grande do Sul, Brazil
[2]Universidade Estadual do Norte Fluminense Darcy Ribeiro, Campos dos Goytacazes, Rio de Janeiro, Brazil
[3]Departamento de Ecologia e Zoologia, Universidade Federal de Santa Catarina, Florianópolis, Santa Catarina, Brazil
[4]Australian Institute of Marine Science, Crawley, Western Australia, Australia
[5]Oceans Institute, The University of Western Australia, Crawley, Western Australia, Australia
[6]Departamento de Oceanografia e Limnologia, Universidade Federal do Rio Grande do Norte, Natal, Rio Grande do Norte, Brazil
[7]Departamento de Biologia Marinha, Universidade Federal Fluminense, Niterói, Rio de Janeiro, Brazil
[8]Centro de Biologia Marinha, Universidade de São Paulo, São Sebastião, São Paulo, Brazil
[9]Departamento de Oceanografia e Ecologia, Universidade Federal do Espírito Santo, Vitória, Espírito Santo, Brazil
[10]Department of Evolution, Ecology and Organismal Biology at The Ohio State University USA

*Correspondence to*: Cesar A.M.M. Cordeiro (cammcordeiro@gmail.com)

**Abstract.** The Brazilian marine biogeographical province (SW Atlantic) hosts coral and rocky reefs that cover ~27 degrees of latitude and are distributed along a relatively narrow continental shelf and four oceanic islands and archipelagos. The broad gradients in temperature, productivity and salinity shape patterns of biodiversity and lead to distinct local communities within the province. Although existing research has helped to unveil spatiotemporal patterns of marine diversity in this province, data availability and scale have limited broader inferences on the main processes shaping biodiversity. Here, we bring together 16 datasets (n = 11 for reef fish, n = 5 for benthic reef organisms) comprising 22 years of research conducted across most of the Brazilian province. These datasets are unprecedented in terms of temporal, spatial, and taxonomic coverage. For example, eight datasets (six for reef fish, two for benthos) span seven (fish monitoring in Rio Grande do Norte) to 18 years (fish monitoring in Arraial do Cabo, Rio de Janeiro) of survey data. Also, these datasets contain data collected in priority areas for conservation in the Brazilian province, such as the Abrolhos Bank and the Trindade island. The data comprise detection and fish count/benthic cover data for 24,498 sampling events deployed at 55 locations, formatted according to the Darwin Core Standard, being therefore interoperable with other existing datasets. The 11 fish datasets comprise the detection and counting of 361 fish taxa (312 identified at species level, 49 identified at genus, subfamily and family) from 178 genera, 71 families and 2 classes (Teleostei and Elasmobranchii). The five benthic datasets comprise the description of the detection and cover of 81 taxa, 82 genera, 68 families, 15 classes, and 4 kingdoms (Animalia, Bacteria,





Plantae, Chromista). By making this an open-access resource, we share with the public the result of two decades of federal

and state funding for scientific research on Brazilian reefs.

**Short summary.** The Brazilian marine biogeographical province features coral and rocky reefs spanning 27º latitude along a narrow continental shelf and four oceanic islands and archipelagos. Our reef synthesis working group reunite 16 datasets (11 on reef fish, 5 on benthos) comprising 24,498 sampling events across 55 locations over up to 22 years. This data offers unparalleled temporal, spatial, and taxonomic coverage, facilitating comprehensive inferences on the processes shaping

biodiversity in the province.

## 1 Introduction

Ecology has been experiencing a transition from isolated research, with data stored locally, to a global research network with data stored in public repositories and promptly available to foster further research (Reichman et al., 2011). As a consequence, the field has experienced a hike in multidisciplinary research to understand patterns and formulate general

principles (Michener and Jones, 2012), and in doing so it has been increasingly promoting the principles of open research and synthesis science (Braga et al., 2023; Reichman et al., 2011; UNESCO, 2021). Brazil started to follow this trend recently. For example, the 'SinBiose', the Synthesis Center on Biodiversity and Ecosystem Services (see http://www.sinbiose.cnpq.br/web/sinbiose/home), is a unique ecological synthesis initiative based in the Global South and world tropics, and fosters innovative ideas about data synthesis, application of open-science principles, and interdisciplinary

and transdisciplinary research (Luza et al., 2023a). The 'SinBiose' program was created by the Brazilian National Council for Scientific and Technological Development (CNPq; www.cnpq.br) to fund initiatives that could synthesize biodiversity data to understand ecosystem services provided by several Brazilian biomes, and therefore better inform their management. Seven working groups were funded in the first SinBiose call, with ReefSYN—Reef Synthesis Working Group—being the only one developing research on marine ecosystems, particularly on reefs distributed within the Brazilian marine

biogeographical province (sensu (Briggs, 1974)). The ReefSYN brings together 22 researchers (20 early-career and senior researchers, assisted by two post-doctoral researchers) from 14 institutions located in Brazil and two other countries (Fig. 1). Funding from 'SinBiose' (CNPq) has given the ReefSYN group the opportunity to unite scientists of different areas of expertise, assemble data, and develop ecological synthesis with a common objective: to investigate the patterns and drivers of reef biodiversity and the provision of reef ecosystem services in Brazil (see https://reefsyn.weebly.com/home-us.html).

The Darwin Core Standard (DCS) provides a common, interoperable and globally accepted protocol for curating datasets. The data formatted following DCS consists primarily of the geographic position of sampled taxa, associated to their occurrence (detection and non-detection) and/or abundance (counts, cover) in locations and localities, as documented by observations, samples, and related information. The metadata associated with the DCS follows a globally adopted glossary of terms (https://dwc.tdwg.org/) intended to facilitate the sharing of information about biological diversity by providing





identifiers, labels, and definitions for compiled data. Adequate data standardization and storage enable data and metadata to be findable, accessible, interoperable and reusable (the 'FAIR' principles, (Wilkinson et al., 2016)).

By adopting the DCS, synthesis initiatives are able to answer ambitious questions about drivers of biodiversity and functioning of Brazilian reef ecosystems, assessing anthropogenic impacts on reef biodiversity and ecosystem services, and incorporating data from the Brazilian province into global assessments. Furthermore, data can be integrated with pre-existing

datasets (e.g., the Ocean Biodiversity Information System, https://obis.org/) by researchers and/or machine-learning algorithms to achieve larger generalities with agility, greater power of inference and with fewer limitations than smaller datasets. Following these principles, we hereby present and characterize 16 datasets (11 for fish and five for benthos) gathered by the ReefSYN team and its collaborators. We detail the data structure, and show summaries of spatial distribution, sampling effort and methods, as well as the accumulation of data over time, and their taxonomic coverage.


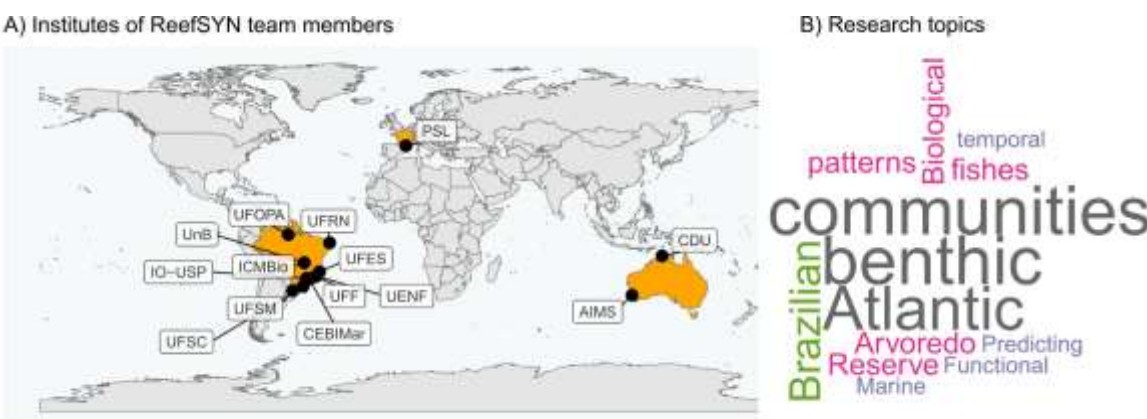

**Figure 1: Institutes (A) and research topics (B) explored by the Reef Synthesis Working Group (ReefSYN). Abbreviations: AIMS: Australian Institute of Marine Science; CDU: Charles Darwin University; CEBIMar: Centre for Marine Biology of the University of São Paulo; ICMBio: Chico Mendes Institute for Biodiversity Conservation;**
**IO-USP: Oceanographic Institute of the University of São Paulo; PSL: Paris Sciences et Lettres University; UENF: North Fluminense State University; UFES: Federal University of Espírito Santo; UFF: Fluminense Federal University; UFOPA: Federal University of Western Pará; UFRN: Federal University of Rio Grande do Norte; UFSC: Federal University of Santa Catarina; UFSM: Federal University of Santa Maria; UnB: University of Brasília.**

**2 Methodology**

**2.1 Geographical and temporal coverage**

The Brazilian marine biogeographical province (Briggs, 1974; Floeter et al., 2008; Pinheiro et al., 2018), located in the Southwestern Atlantic, hosts reefs with mostly turbid and nutrient-rich waters due to the sediment discharge from several rivers on the coast (Aued et al., 2018; Loiola et al., 2019; Mies et al., 2020). Recent analysis of spatially extensive benthic datasets revealed clear-water reef communities occurring in the oceanic islands and in more oligotrophic waters, whereas



turbid-water reef communities occur along most of the coast (Santana et al., 2023). Biogenic and rocky reefs are distributed along approximately 27 degrees of latitude (0.91N to 27.6S latitude degrees; Fig. 3) and exposed to different temperature, productivity and salinity regimes, which generates a pronounced regionalization of the biodiversity (Cord et al., 2022; Luza et al., 2023b; Pinheiro et al., 2018). Four oceanic islands are located off the Brazilian continental shelf, which host coralligenous and rocky reefs, with high endemism levels, and a subset of the species composition of coastal reefs (Cord et

al., 2022; Pinheiro et al., 2018).

We compiled data collected between 2001 and 2023 on fish and benthic organisms (e.g., algae, corals) from 55 locations and 355 unique localities (n=317 unique localities for fish, n=138 unique localities for benthos) distributed along the Brazilian coast and oceanic islands (Fig. 2, Tables 1 and 2). 'Locality' defines a set of replicates of transects at a given time and place, and 'location' defines a broad set of localities within a region or island. All these data came from geographically replicated,

large-scale and long-term ecological monitoring research programs conducted over the last decades in Brazil (Sisbiota-Mar, PELD-ILOC, Abrolhos Bank monitoring), and from novel initiatives such as the monitoring of reef fish and benthos of Rio Grande do Norte (Roos et al., 2019) and São Paulo states (Barreto et al., in prep.).

The datasets encompass information gathered from eight key conservation areas within the Brazilian marine province, identified by the codes ZCM-91, ZCM-108, ZCM-109, ZCM-122, ZCM-134, ZCM-51, ZCM-87, ZCM-88, ZCM-46, ZCM-

78 and ZCM-53 (Ministério do Meio Ambiente, 2018). These areas include the Abrolhos Bank, Trindade island, Central-North Santa Catarina, coastal islands of São Paulo (Santos to Ubatuba), Central-South Espírito Santo, Ilha Grande bay, and the Environmental Protection Area (Área de Proteção Ambiental, APA) Recifes de Corais in the state of Rio Grande do Norte. Out of the 317 unique localities in the fish datasets, 47% (n=149) coincide with these priority conservation areas. Similarly, among the 138 unique localities in the benthic datasets, 48% (n=66) are situated within these priority zones.

The sampling effort available in the current datasets was not evenly distributed (Fig. 3), resulting in a varied accumulation of taxonomic information over time (Fig. 4). The cumulative number of eventIDs (i.e., information associated with a sampling event—i.e., something that occurs at a place and time—) over time and across the datasets show that: i) there was an abrupt increase in the number of eventIDs after 2013-2014, mainly for benthos (leveraged by the Sisbiota-Mar project (CNPq)); ii) large projects such as Sisbiota-Mar and PELD-ILOC have built on previous initiatives (Krajewski and Floeter, 2011;

Pinheiro and Gasparini, 2009); iii) there was a nearly constant increase in the number of fish taxa over time. On the other hand, for benthos, up to 2014 the sampling effort remained relatively stable and limited over time, followed by an abrupt increase in the number of sampling events and taxa (Figs. 3 and 4). This pattern is probably caused by the laboratory and computational work required to identify these organisms into finer taxonomic scales. The prominent increase in sampling effort after 2012–2013 is associated with the Sisbiota-Mar network, the increase in collaboration networks, aggregation of

datasets, and the total amount of financial support derived from network cooperative efforts/partners. The ProspecMar project (CNPq), for example, which is primarily focused on bioprospection and holobiotic investigation, has been an important contribution to maintain fieldwork in the oceanic islands after 2013.

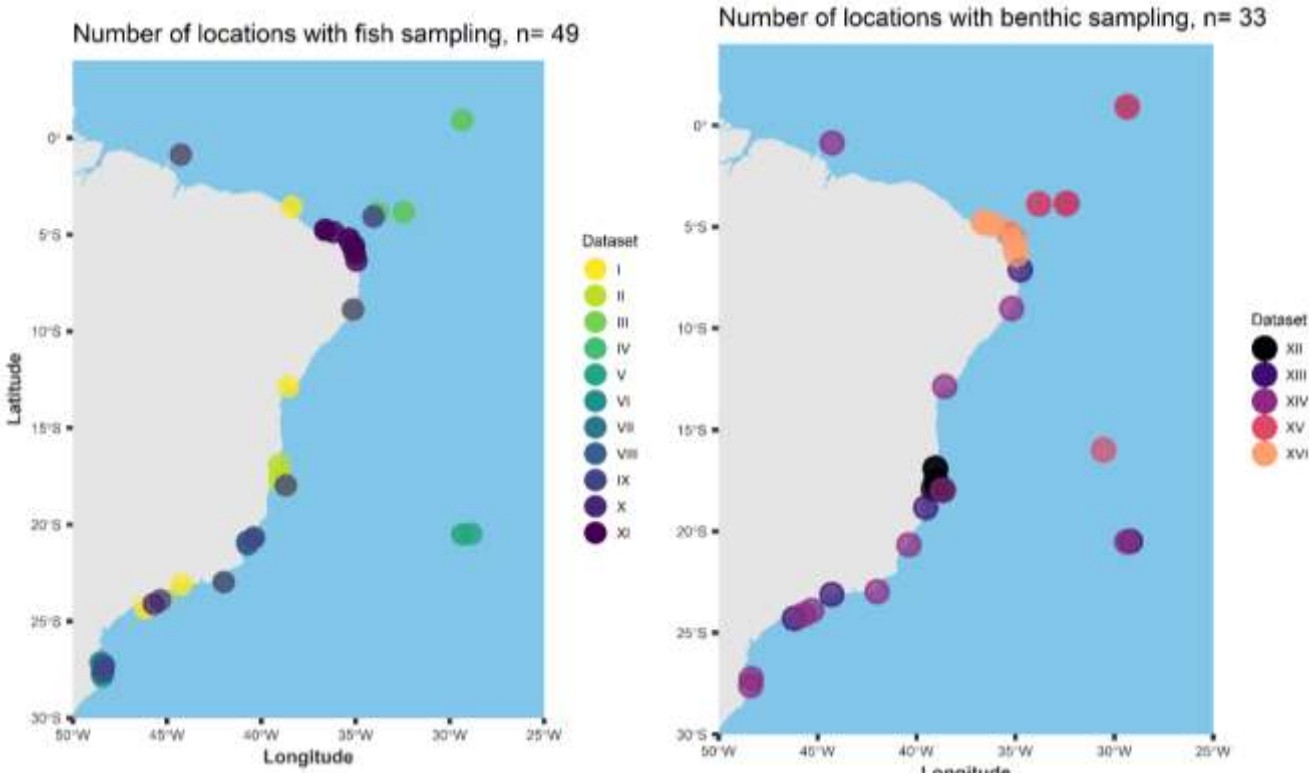

**Figure 2: Spatial distribution of fish (left) and benthos (right) sampling locations per dataset. Datasets: I - Fish communities from the Brazilian province, II - Abrolhos Bank monitoring, III - Arraial do Cabo (Rio de Janeiro) monitoring, IV - Oceanic islands' monitoring, V - Fish assemblages from Trindade and Martin Vaz, VI - Santa Catarina monitoring, VII - Fish assemblages from Guarapari, Espírito Santo, VIII - Fish assemblages from Southern Espírito Santo, IX - Trophic interactions along the Western Atlantic, X - Alcatrazes monitoring, XI - Rio Grande do Norte monitoring, XII - Benthic communities' monitoring in Abrolhos Bank, XIII - Extended benthic communities from the Brazilian province, XIV - Benthic communities from the Brazilian province, XV - Benthic communities' monitoring from oceanic islands, XVI - Benthic communities from Rio Grande do Norte.**

### 2.2 Data description

The 16 datasets described here represent different efforts to gather data on reef fish and benthos along the Brazilian biogeographical province. Six out of the 11 fish datasets and two out of five benthic datasets are time series (Tables 1 and 2). The remaining datasets are spatial snapshots (only one visit to a locality) through which data were collected on different events over many years. The 16 datasets include a total of 24,498 samples of fish and benthos, distributed throughout the Brazilian biogeographic province (n = 20,561 for fish, and n = 3,937 for fish) and collected from 2001 to 2023 (fish: 2001 - 2023, benthos: 2003 - 2019) (Fig. 2). These samples were deployed at a total of 55 locations and 355 localities, and involved 91 different observers. The citation and the DOI of each data set is provided in the Table 3.





Open Access — Earth System Science Data Discussions

**Table 1: Description of the eleven fish datasets regarding sampling protocol, number of sampling events and geographic and temporal range. Sampling unit area is provided for each transect/underwater visual census.**

| Dataset/Name | Sampling Protocol | Samples (n) | Scale | Sampling unit area (m²) | Number of locations | Latitude | Longitude | Years |
|---|---|---|---|---|---|---|---|---|
| I — Fish communities from the Brazilian province | Underwater visual survey - 20 × 2m | 4570 | Transect/plot | 40 | 20 | -27.85, 0.92 | -48.52, -28.86 | 2001-2015 |
| II — Abrolhos Bank monitoring | Stationary visual survey - 4 × 2m | 6422 | Transect/plot | 8 | 5 | -18, 16.89 | -39.15, 38.65 | 2001-2014 |
| III — Arraial do Cabo (Rio de Janeiro) monitoring | Underwater visual survey - 20 × 2m | 2146 | Transect/plot | 40 | 1 | -23.01, 22.96 | -42.04, 41.98 | 2003-2021 |
| I — Oceanic islands' monitoring | Underwater Visual Survey - 20 × 2m | 3561 | Transect/plot | 40 | 4 | -20.53, 0.97 | -33.82, 28.86 | 2006-2019 |
| V — Fish assemblages from Trindade and Martin Vaz | Underwater visual survey - 20 × 2m | 344 | Transect/plot | 40 | 2 | -20.53, 20.47 | -29.35, 28.86 | 2007-2009 |
| V — Santa Catarina monitoring | Underwater visual survey - 20 × 2m | 1897 | Transect/plot | 40 | 9 | -27.84, 27.12 | -48.53, 48.33 | 2007-2021 |
| V — Fish assemblages from Guarapari, Espírito Santo | Underwater visual survey - 20 × 2m | 320 | Transect/plot | 40 | 1 | -20.70, 20.68 | -40.41, 40.36 | 2008-2008 |
| V — Fish assemblages from Southern Espírito Santo | Underwater visual survey - 20 × 2m | 255 | Transect/plot | 40 | 5 | -21.14, 20.83 | -40.79, 40.57 | 2009-2009 |





**145** **Table 2. Description of the five benthic datasets regarding sampling protocol, number of sampling events and geographic and temporal range. Sampling unit area is provided for each photoquadrat.**

| Dataset/Name | Sampling Protocol | Samples (n) | Scale | Sampling unit area (m²) | Number of locations | Latitude | Longitude | Years |
|---|---|---|---|---|---|---|---|---|
| I Trophic interactions along the Western Atlantic | Video plot - 2 × 1m | 776 | Transect/plot | 2 | 12 | -27.6, -0.87 | -48.39, -34.04 | 2009-2014 |
| X Alcatrazes | Underwater visual survey - 20 × 2m | 329 | Transect/plot | 40 | 1 | -24.11, -24.1 | -45.71, -45.69 | 2013-2022 |
| X Rio Grande do Norte monitoring | Underwater visual survey - 20 × 2m | 859 | Transect/plot | 40 | 7 | -6.38, -4.72 | -36.7, -34.93 | 2016-2023 |

| Dataset/Name | Sampling Protocol | Samples (n) | Scale | Sampling unit area (m²) | Number of locations | Latitude | Longitude | Years |
|---|---|---|---|---|---|---|---|---|
| XII Benthic communities' monitoring in Abrolhos Bank | Photoquadrats - 0.5 × 0.5 m | 54 | Plot/point | 0.25 | 5 | -17.91, -16.89 | -39.15, -38.94 | 2003-2005 |
| XIII Extended benthic communities from the Brazilian province | Photoquadrats - 0.66 × 0.75 m | 595 | Plot/point | 0.49 | 4 | -17.98, -17.46 | -39.03, -38.66 | 2006-2014 |
| XIV Benthic communities from the Brazilian | Photoquadrats - 0.66 × 0.75 m | 24 | Plot/point | 0.50 | 7 | -24.29, 0.92 | -46.18, -28.86 | 2008-2018 |
| | Photoquadrats - 0.25 × 0.25 m | 771 | Plot/point | 0.06 | 15 | -27.6, 0.87 | -48.39, -29.31 | 2010-2014 |





150

| Dataset/Name | Sampling Protocol | Samples (n) | Scale | Sampling unit area (m²) | Number of locations | Latitude | Longitude | Years |
|---|---|---|---|---|---|---|---|---|
| province | | | | | | | | |
| XV | Benthic communities' monitoring from oceanic islands | Photoquadrats 0.5 × 0.5 m | 2748 | Plot/point | 0.25 | 4 | -20.52, - 0.92 | -33.82, - 29.32 | 2013- 2019 |
| XVI | Benthic communities from Rio Grande do Norte | Photoquadrats - 0.25 × 0.25 m | 285 | Plot/point | 0.06 | 7 | -6.38, - 4.75 | -36.69, - 34.93 | 2016- 2017 |

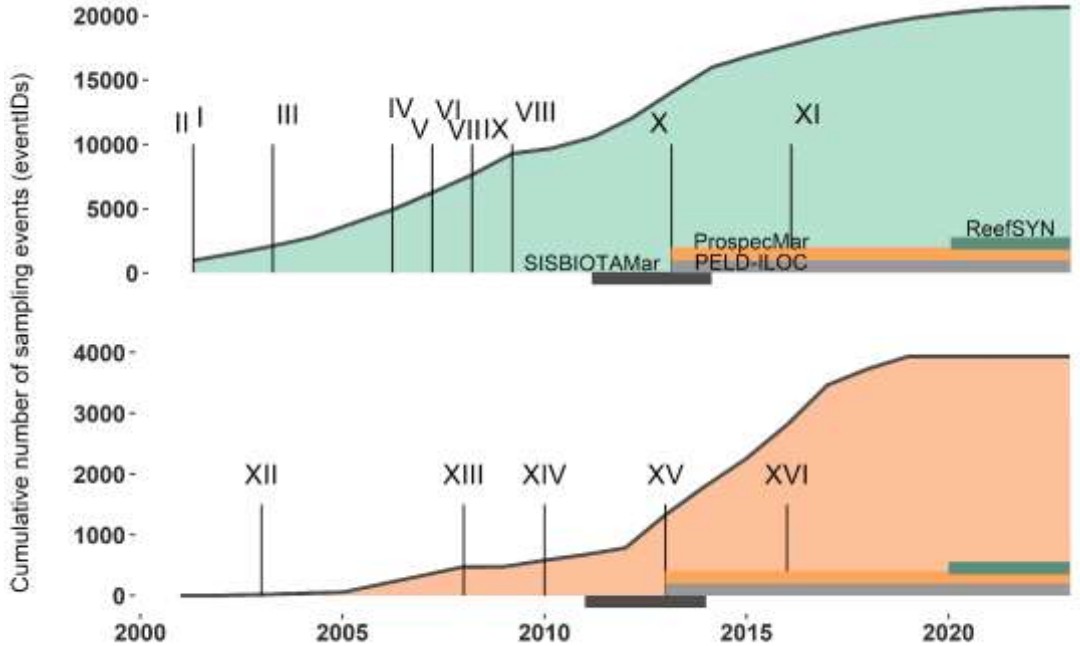

**Figure 3: The number of sampling events accumulated over time for fish (top) and benthos (bottom). One sampling event consists, for example, on one transect deployed into a locality. The vertical segments indicate the year in which data included in each dataset started to be collected. The horizontal bars depict the main funding sources as follows: dark gray bar: Sisbiota-Mar, CNPq; light gray bar: PELD ILOC, CNPq; orange bar: ProspecMar-Ilhas, CNPq; green bar: ReefSYN, SinBiose CNPq. Datasets: I - Fish communities from the Brazilian province, II - Abrolhos Bank monitoring, III - Arraial do Cabo (Rio de Janeiro) monitoring, IV - Oceanic islands' monitoring, V - Fish assemblages from Trindade and Martin Vaz, VI - Santa Catarina monitoring, VII - Fish assemblages from Guarapari, Espírito Santo, VIII - Fish assemblages from Southern Espírito Santo, IX - Trophic interactions along the Western Atlantic, X - Alcatrazes monitoring, XI - Rio Grande do Norte monitoring, XII - Benthic communities' monitoring in Abrolhos Bank, XIII - Extended benthic communities from the Brazilian province, XIV - Benthic communities from the Brazilian province, XV - Benthic communities' monitoring from oceanic islands, XVI - Benthic communities from Rio Grande do Norte.**

Earth System
Science
Data

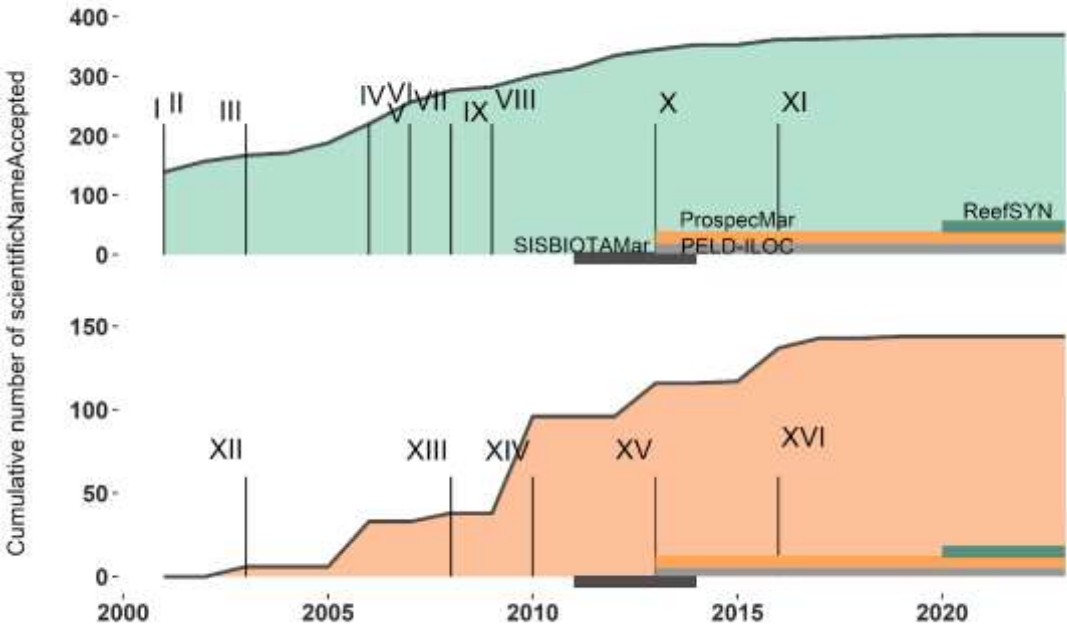

**Figure 4: Trends in the number of scientific names (accepted names, at any taxonomic level) accumulated over time for fish (top) and benthos (bottom). The vertical segments depict the year in which data included in each dataset started to be collected. The horizontal bars depict the formal funding as follows: dark gray bar: Sisbiota-Mar, CNPq; light gray bar: PELD ILOC, CNPq; orange bar: ProspecMar-Ilhas, CNPq; green bar: ReefSYN, SinBiose CNPq. Datasets: I - Fish communities from the Brazilian province, II - Abrolhos Bank monitoring, III - Arraial do Cabo (Rio de Janeiro) monitoring, IV - Oceanic islands' monitoring, V - Fish assemblages from Trindade and Martin Vaz, VI - Santa Catarina monitoring, VII - Fish assemblages from Guarapari, Espírito Santo, VIII - Fish assemblages from Southern Espírito Santo, IX - Trophic interactions along the Western Atlantic, X - Alcatrazes monitoring, XI - Rio Grande do Norte monitoring, XII - Benthic communities' monitoring in Abrolhos Bank, XIII - Extended benthic communities from the Brazilian province, XIV - Benthic communities from the Brazilian province, XV - Benthic communities' monitoring from oceanic islands, XVI - Benthic communities from Rio Grande do Norte.**

## 2.3 Data sources and sampling protocol

### 2.3.1 Dataset I: Fish communities from the Brazilian province

This dataset, an updated, filtered and curated version of the one used by (Morais et al., 2017), includes fish counts and size estimates in 4,570 transects distributed over 137 localities in 20 different locations, spanning from 0° to 27°S (including oceanic islands). Sampling descriptors include Observer ID, locality depth, and date. The geographical information (coordinates) is indicated here at the locality level. Fish were tallied via underwater visual census (UVC) along $20 \times 2$ m belt transects, and samples were obtained in the austral summer from 2001 to 2015. Strip transects performed by snorkelling or



scuba diving, during which the diver unwound a tape while identifying, tallying and visually estimating the total length (TL, cm) of non-cryptic fishes >10 cm. While retracting the tape, the diver followed the same procedure for benthic-associated non-cryptic fishes <10 cm and cryptic species (see Morais et al., 2017 for more details). The dataset also includes data from
Krajewski and Floeter (2011). Data on individuals and their lengths, together with metadata, are provided in the Darwin Core Standard terms "measurementValue", "measurementType", and "measurementUnit" for all fish datasets. This dataset is published in (Morais et al., 2024).

### 2.3.2 Dataset II: Abrolhos Bank monitoring

Fish assemblage data from the Abrolhos Bank was collected by several collaborators under the coordination of Ronaldo B.
Francini-Filho from 2001 to 2014. The data was obtained with support from Conservation International (CI) and FAPESP between 2001 and 2005, and exclusively by CI from 2006 to 2009, within the Marine Management Areas Science Program. Between 2010-2014, RBFF continued to coordinate the Abrolhos monitoring program within the Abrolhos-SISBIOTA project (under general coordination of Dr. Alex Bastos) (see (Francini-Filho et al., 2013, 2019; Francini-Filho and De Moura, 2008; Roos et al., 2020)). This dataset includes samples of five locations and 28 localities nested within locations,
but not evenly distributed in space. The variables in this dataset are related to the description of fish identification, size and counts, as estimated with nested stationary visual censuses ($4 \times 2$ m, 5 min; cf. (Minte-Vera et al., 2008)). Depth of sampling varied among localities and ranged between 2 and 15 m. Due to its large size, the published data set does not include absences (please contact the authors to access these data). This dataset is published in (Francini-Filho, 2024a).

### 2.3.3 Dataset III: Arraial do Cabo (Rio de Janeiro) monitoring

The data were collected by the Reef Systems Ecology and Conservation Lab (LECAR) team (https://www.lecar.uff.br/) from 2003 to 2021, but were annually collected only from 2014 to 2021 at four of the 21 localities monitored. All other localities were sampled opportunistically. Dataset includes fish species identification, size and counts recorded at 40 m² ($20 \times 2$ m) belt transects—similarly to what was described for Dataset I and used by Morais et al. (2017)—in rocky reefs in Arraial do Cabo, Rio de Janeiro. Transects were laid at different depths at approximately 5 m intervals according to local maximum
depths, which ranged from 1 to 25 m. Samples include data from two distinct oceanographic domains that are simultaneously present in Arraial do Cabo, one under strong influence of seasonal upwelling, and another which is indirectly influenced by the upwelling. This dataset (Ferreira et al., 2024) is embargoed up until January 2025 (see **Disclaimer** and Table 3).

### 2.3.4 Dataset IV: Oceanic islands' monitoring

Dataset of fish recorded in the four oceanic islands of Brazil: Fernando de Noronha Archipelago, Rocas' Atoll, Trindade
Island and Martin Vaz Archipelago, and Saint Peter and Saint Paul's Archipelago. Data were collected from 2013 to 2023, organized by Juan P. Quimbayo, Thiago Silveira and Cesar Cordeiro (PELD-ILOC team, https://peldiloc.sites.ufsc.br/) and



curated by Cesar Cordeiro. As the data from 2020 to 2022 are still under processing, they were not included here. Fish had their identification, size and count tallied via the underwater visual census (UVC). These data were generated by the team of PELD ILOC project applying the same UVC protocol (20 × 2 m belt transect) described above and used by Morais et al.

(2017). Transects were laid at different depths at 5 m intervals according to local maximum depths, which ranged from 3 to 25 m. In this dataset, "island" name equals "location" name in other datasets (Fig. 5). This dataset is published in (Cordeiro et al., 2021)

### 2.3.5 Dataset V: Fish assemblages from Trindade and Martin Vaz

The data were collected by Hudson T. Pinheiro during two expeditions to Trindade Island, organized by the TAMAR-

ICMBio Project and the Brazilian Navy. The expeditions occurred between February and April 2007, and April and July 2009, resulting in 120 days in the field and over 100 SCUBA dives. The data in Martin Vaz was collected in the 2007 expedition, during a three-days trip aboard on a fishing boat. The UVCs were conducted on 20 x 2 m belt transects and covered fringing and rocky reefs, different levels of substrate complexity and were performed between 3 and 45 m depth (see Pinheiro 2011 for details). Fishes were tallied within size classes (0-5 cm, 5-10, 10-20 and so on) to estimate biomass

through length-weight relationships. The results of the study were published in (Pinheiro and Gasparini, 2009), (Pinheiro et al., 2009, 2011), and (Simon et al., 2013a). This dataset is published in (Pinheiro, 2024).

### 2.3.6 Dataset VI: Santa Catarina monitoring

The data was collected yearly by the LBMM team (https://lbmm.ufsc.br/), from 2007 to 2022. Dataset includes fish species, size and abundance recorded at 40 m² (20 × 2 m) belt transects at nine locations along the Santa Catarina state coastal area.

Transects were laid at different depths at approximately 5 m intervals according to local maximum depths, which ranged from 1 to 25 m. This dataset, named TimeFISH, was recently published by (Quimbayo et al., 2022), and a Darwin Core version is published in (Quimbayo et al., 2024).

### 2.3.7 Dataset VII: Fish assemblages from Guarapari, Espírito Santo

The data were collected by Thiony Simon (in memoriam) and Hudson T. Pinheiro for the former's MSc thesis

(https://ictiolab.wordpress.com), and the results of the study were published in (Simon et al., 2011, 2013a, b). The data was SCUBA-collected between January and March 2008 at four locations close by to each other, and located about ten kilometers offshore Guarapari, Espírito Santo, central coast of Brazil. Two locations referred to natural habitats, the Rasas islands (113 UVCs; depth 3 to 25 m) and the island Escalvada (126 UVCs; 3-25 m). The other two locations were the shipwreck Bellucia (35 UVCs; 20-27 m), that ran aground and sank in 1903, and the derelict vessel Victory (46 UVCs; 18-

35 m) that was intentionally sunk in 2003 to serve as an artificial reef. The 324 UVCs were conducted on 20 x 2 m belt transects as described earlier (see Morais et al. (2017) for more details). Fishes were tallied within size classes (0-5 cm, 5-10,



10-20 and so on) to estimate biomass through length-weight relationships. Funding by FAPES. This dataset is published in (Pinheiro and Simon, 2024a)

### 2.3.8 Dataset VIII: Fish assemblages from Southern Espírito Santo

The data were collected by Hudson T. Pinheiro and Thiony Simon (in memoriam), with support of the IctioLab and Lab Necton UFES groups. The study area covers the southern coast of the state of Espírito Santo (between the municipalities of Anchieta and Mataraízes), between latitudes 20°42'S and 21°09'S. The data was used in a systematic spatial planning for marine conservation, resulting in a proposal for the creation of a mosaic of marine protected areas in the region (Pinheiro et al., 2010; Teixeira et al., 2013). More recently, the data was used for the thesis of Guilherme L. da Cruz and is currently
under review (Cruz et al., under revision). The data was SCUBA-collected in the summer of 2009 and consists of 251 UVCs (20 x 2 m transects) performed between 3 to 23 meters depth. Divers registered the main substrate features in all UVCs and characterized five main habitats: biogenic reefs, rocky reefs, rhodoliths with algae, rhodoliths with invertebrates, and rhodoliths with sand. This dataset is published in (Pinheiro and Simon, 2024b)

### 2.3.9 Dataset IX: Trophic interactions along the Western Atlantic

This dataset, used by (Longo et al., 2019) and (Inagaki et al., 2020), includes records of feeding behavior of fish over the benthic communities, as well as interactions among fish. These data were obtained from 1,133 unique videoplots deployed in 70 localities from 17 different locations spanning 61 degrees of latitude, from 34°N to 27°S. Sampling descriptors include recording time, date, depth, and observed ID. At each locality, static videos were replicated at $2 \times 1$ m areas positioned haphazardly on the reefs, with 5–10 m between replicates. Fish not identified in the videoplots (coded as 'not_identified')
were removed from the dataset. Feeding pressure was estimated as the product of the number of bites taken and the body mass (in grams) of the fish, accounting for body size variation (sensu (Longo et al., 2014)). Individual biomass was obtained through length–weight relationships from FishBase. Although we describe here the Brazilian Province subset, the complete dataset is published on OBIS (Longo and Inagaki, 2024).

### 2.3.10 Dataset X: Alcatrazes monitoring

This dataset has been collected through the collective effort among Instituto Chico Mendes, Centro de Biologia Marinha, Universidade de São Paulo (CEBIMar/USP), LECAR and LBMM teams. The dataset includes data from underwater visual census (UVC) performed along $20 \times 2$ m (40 m²) belt transects from 1.5 to 17 m deep. The UVCs were performed using the same protocol described above and used by Morais et al. (2017). Three different localities with varying protection levels within an MPA were monitored between 2013 and 2022. This dataset is published in (Mendes et al., 2024).





### 2.3.11 Dataset XI: Rio Grande do Norte monitoring

Data were collected by Guilherme Longo and Natália Roos in Rio Grande do Norte state, northeastern Brazil. The dataset includes fish species, size, abundance and biomass recorded at 40 m² (20 × 2m) belt transects during UVC surveys. A total of 820 UVCs were performed at 54 localities within 7 different locations at depths ranging from 1 to 28 m depending on local depth. Data has been collected yearly since 2016. During the first year, only fish from clades Scarini and Acanthuridae were tallied. From 2017 on, the entire fish assemblage began to be surveyed. These two first years accounted for 320 UVCs conducted at all 7 locations, most of them surveyed just once in time (Roos et al., 2019). Two localities within the shallow patchy reefs from the municipality of Rio do Fogo kept being monitored every four months from April 2018 to April 2023 (onwards) by Jessica Bleuel and other members of the Marine Ecology Laboratory Team (https://twitter.com/MarineEcoBR), adding up more than 850 UVCs. This dataset is published in (Longo et al., 2024).

### 2.3.12 Dataset XII: Benthic communities' monitoring in Abrolhos Bank

Benthic community assessments were performed at the same locations, localities and depths as fish stationary censuses (Dataset II), using either point-intersect technique (four 10- m -transects in each depth and locality, with organisms identified in situ) or photoquadrats (~10 quadrats, 0.49 m²; the sample size (effort) varies across locations and years (Francini-Filho et al., 2013)) (Table 2). For the photoquadrat method, a mosaic of 15 high-resolution digital images totalling 0.3 m² constituted each sample (Francini-Filho et al., 2013). Quadrats were haphazardly set between 2001-2007 along a 20–50 m axis on the tops and walls of reef pinnacles and permanently delimited by fixed metal pins from 2008-2014. Relative coral cover was estimated through the identification of organisms below 300 randomly distributed points per quadrat (i.e., 20 points per photograph) using the Coral Point Count with Excel extensions software (CPCe v.4.1) (Kohler and Gill, 2006). The counts of benthic organisms (i.e. random points) were converted to percentages. The cover of benthic organisms and associated metadata are present in the DCS terms "measurementValue", "measurementType", and "measurementUnit" for all benthic datasets. This dataset is published in (Francini-Filho, 2024b).

### 2.3.13 Dataset XIII: Extended benthic communities from the Brazilian province

This dataset was compiled by Erika Santana, Anaide W. Aued, and Ronaldo Francini-Filho, and consists of data on benthic organisms sampled in photoquadrats deployed in several locations disposed along the Brazilian coast and oceanic islands (see (Santana et al., 2023)). This dataset is complementary to the dataset of Aued et al. (2018). Image processing was done using the Coral Point Count with Excel extensions software (CPCe v. 4.1) (Kohler and Gill, 2006). Benthic organisms were identified at different taxonomic levels (morphotype, species, order). Morphotypes were adapted from (Steneck and Dethier, 1994) whereby algae are grouped according to morpho-anatomical characteristics. However, morphotype, bare substrate, sediment, lost information (shade, quadrat, tape) and turf were not included in the data because they do not represent





taxonomic entities in which DCS standards are based. The dataset originally had environmental descriptors such as depth, month and year. This dataset was combined with that of (Aued et al., 2018) in the analysis of (Santana et al., 2023). This dataset is published in (Santana et al., 2024).

### 2.3.14 Dataset XIV: Benthic communities from the Brazilian province

This dataset, used by (Aued et al., 2018)—data also published on DRYAD
(https://datadryad.org/stash/dataset/doi:10.5061/dryad.f5s90) —includes plot-level cover information of ~100 benthic taxa from 3,855 photoquadrats deployed at 40 localities from 15 different locations, spanning 0° to 27°S. The sampling localities indicated here are the same from Morais et al. (2017). Benthic organisms were identified at the lowest possible taxonomic level (i.e., morphotype, species, order) according to constraints related to image identification. Image processing was done using the Coral Point Count with Excel extensions software (CPCe v. 4.1) (Kohler and Gill, 2006) or the photoQuad
software (Trygonis and Sini, 2012). Bare substrate, sediment, lost information (shade, quadrat, tape) and turf were not included in the data because they do not represent taxonomic entities in which DCS standards are based. Sampling descriptors include photoquadrat ID, depth, date or year and, for some samples, observer ID. The geographical information is indicated at the locality level. Six to 20 2 × 1 m horizontal surfaces of reef area on each depth strata, separated by at least two meters of distance from each other, were considered independent sample units. Plots were haphazardly selected for the
photoquadrats (25 × 25 cm; following the sampling design in (Longo et al., 2019) (Dataset IX). This dataset is published in (Aued et al., 2024).

### 2.3.15 Dataset XV: Benthic communities' monitoring from oceanic islands

Dataset of benthic communities recorded in the four oceanic islands of Brazil: Fernando de Noronha Archipelago, Rocas' Atoll, Trindade Island and Martin Vaz Archipelago, and Saint Peter and Saint Paul's Archipelago. These data were collected
from 2013 to 2022, organized by Thiago Silveira and Cesar Cordeiro (PELD-ILOC team) and curated by Cesar Cordeiro. These data were generated by the team of PELD ILOC project (https://peldiloc.sites.ufsc.br/), and are still being sampled annually. As the images from 2020 to 2022 are still under processing, these data were not included here. The method for registering the benthic community included three to six fixed transects (20 m) parallel to the coastline placed, at least, at 2 m intervals. Ten to 11 (50 × 50 cm) photoquadrats were taken at each transect in each year and locality from 2013 to 2019.
Following the imaging register, image processing was done using the Coral Point Count with Excel extensions software (CPCe v. 4.1) (Kohler and Gill, 2006). This stage consisted on the identification of major taxonomic, morpho-anatomical benthic groups and the estimation of their relative cover in samples. Software analysis was performed by overlaying 50 random points on each image and identifying the organisms immediately below each point details (details in (Biscaia Zamoner et al., 2021)). This dataset is published in (Cordeiro et al., 2022).

### 2.3.16 Dataset XVI: Benthic communities from Rio Grande do Norte

This dataset was collected at the same spatial unit from the Dataset XI by Guilherme Longo and Natália Roos. Benthic cover was sampled with photoquadrats of $25 \times 25$ cm every 2 m within the 40 m² ($20 \times 2$ m) belt transects, resulting in 10 photos per transect. Each image was analyzed with the software 'photoQuad' (Trygonis and Sini, 2012) by randomly laying 30 points over the phoquadrat area. The organisms below each point were identified into morpho-functional groups, and to species or genus level when possible. This dataset contains 285 photoquadrats (benthic data gathered around UVCs of dataset VIII) between 2016 and 2017 at seven distinct locations. These data were used in the same publication indicated in the fish monitoring data (Roos et al., 2019). Since then, the Marine Ecology Laboratory Team (https://twitter.com/MarineEcoBR) continued monitoring two localities within the shallow patchy reefs from the municipality of Rio do Fogo with this same method until August 2023. As more images are processed and temporal data are generated, these will be organized accordingly and added to this repository. This dataset is published in (Longo and Roos, 2024).

### 2.4 Data management and standardization

We strived to standardize data to achieve to a Darwin Core standard, and follow the FAIR principles of data science—i.e. data should be "findable", "accessible", "interoperable", and "reusable" (Wilkinson et al., 2016). Data owners supplied their datasets to the database manager in digital format (e.g., spreadsheets "xlsx", "csv"). The datasets were predominantly managed in the R Programming Environment (R Core Team, 2023).

### 2.5 Data structure

All datasets are available as a Darwin Core Archive (DwC-A), and all fields were named compliant with Darwin Core standards that includes an event core (event sampling data), occurrence (taxonomic data), and extended measurements or facts (environmental variables and taxa counts or cover) (Fig. 5).


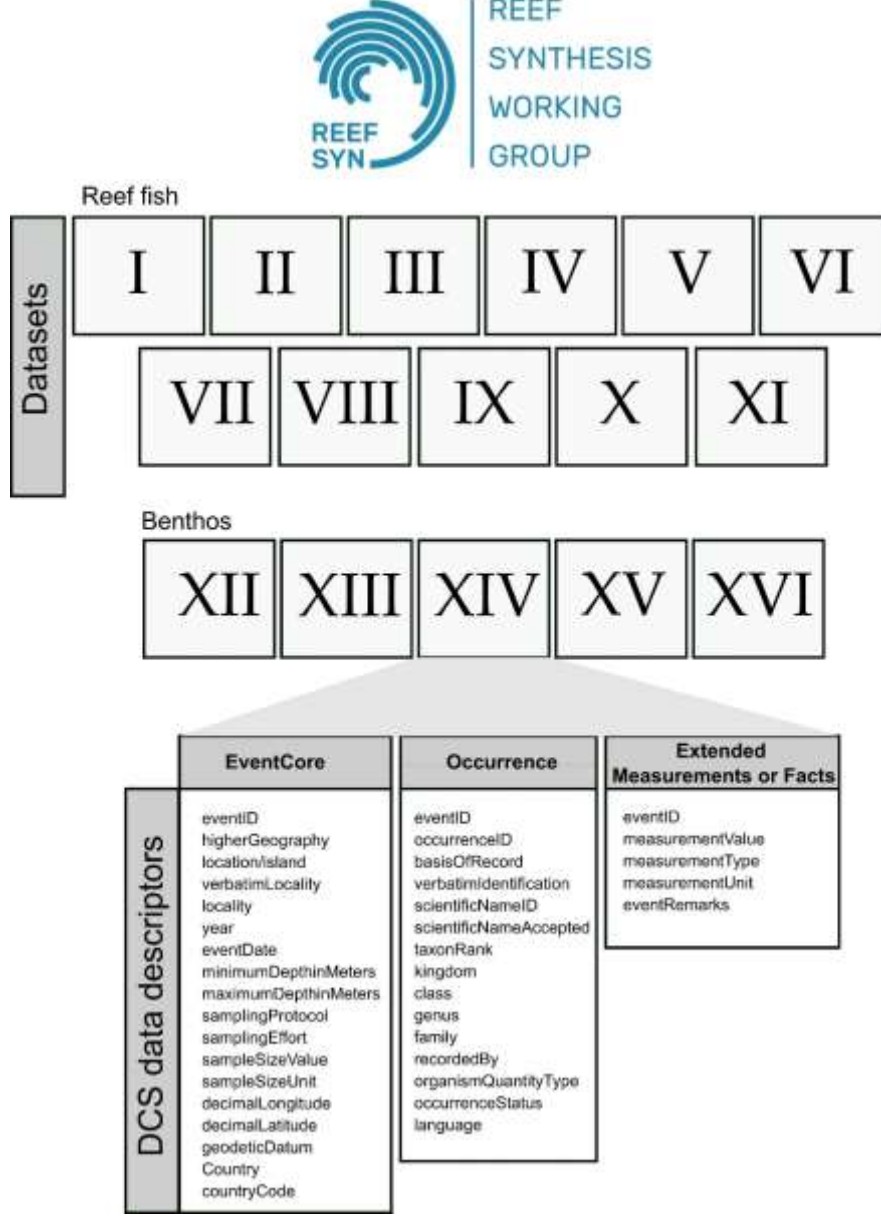

**Figure 5: Structure of the datasets compiled by the ReefSYN working group, showing the Darwin Core Standard (DCS) terms included in most datasets. Some data sets can have additional terms, such as "organismID" in data set IX. These terms are part of a standardized glossary maintained by the Darwin Core Task Group (2009). Datasets: I - Fish communities from the Brazilian province, II - Abrolhos Bank monitoring, III - Arraial do Cabo (Rio de Janeiro) monitoring, IV - Oceanic islands' monitoring, V - Fish assemblages from Trindade and Martin Vaz, VI - Santa Catarina monitoring, VII - Fish assemblages from Guarapari, Espírito Santo, VIII - Fish assemblages from Southern Espírito Santo, IX - Trophic interactions along the Western Atlantic, X - Alcatrazes monitoring, XI - Rio Grande do Norte monitoring, XII - Benthic communities' monitoring in Abrolhos Bank, XIII - Extended benthic communities**



**from the Brazilian province, XIV - Benthic communities from the Brazilian province, XV - Benthic communities'**
**monitoring from oceanic islands, XVI - Benthic communities from Rio Grande do Norte.**

**2.6 Quality assurance/quality control procedures**

● The name of all taxa was checked against the WoRMS database (World Register of Marine Species (WoRMS,
2022)), using the R package "worrms" (Chamberlain and Vanhoorne, 2023). Thus, valid scientific names were called
"scientificNameAccepted" following the DCS standard. The last checking was done on 23 October 2023, using the version
0.4.2 of 'worrms' R package. Some taxonomic updates were made by hand (*Hypanus berthalutzae* (previously *Dasyatis
americana*), *Bathytoshia centroura* (*Dasyatis centroura*), *Azurina multilineata* (*Chromis multilineata*), *Caranx bartholomaei*
(*Carangoides bartholomaei*), *Choranthias salmopunctatus* (*Antias salmopunctatus*), *Goblioclinus kalisherae* (*Labrisomus
kalisherae*), *Sphoeroides camila* (*Sphoeroides spengleri*), *Serranus flaviventris* (*Serranus atricauda*). We decided to include
*Menephorus* (hybrid between *Cephalopholis fulva* and *Paranthias furcifer*) in the dataset so that the record is maintained
even with future taxonomic updates of this organism. Also, Scaridae was considered as part of the Labridae family.

● Samples were always collected by researchers or trained students.

● Sampling methods are broadly used and accepted worldwide.

● Data were checked by two data managers (A.L. Luza, C. Cordeiro) and questions were sent to data owners
whenever necessary. The data owners are listed in the Author contributions' section. Overall, main inconsistencies found in
the datasets (and solved by contacting the data owners) were related with 1) region names; 2) location names; 3) locality
names; 4) different IDs of unique sampling events; 5) format of sampling day, month and year; 6) missing sampling day,
month, year; 7) lack of geographical information; 8) misspelling of species names; 9) sampling unit identity (e.g., one video
plot, photoquadrat).

● Data owners shared Microsoft Excel spreadsheets (".csv", ".xlsx"), often containing data in a wide format. Most
data were transformed into a long format, organized, standardized (following the DCS standard) and processed using the R
Programming Environment (R Core Team, 2023). Some modifications that could not be easily done in R, regarding the
splitting of sample IDs, were done in Microsoft Excel. It consisted in dealing with eventIDs separated by different separators
(" _ ", " . ", …) and with different lengths and content (e.g. some were the concatenation of location, locality, observer, year,
method and sampling ID, whereas others missed some of these components).

● R routines (scripts) are available on GitHub, together with the raw data. R routines will be stored in Zenodo to keep
stable versions of data curation steps. Software used to quantify benthic cover are cited in each dataset description.





## 2.7 Taxonomic coverage

**2.7.1 General taxonomic coverage**

The 11 fish datasets comprise the description of the detection and counting of 361 fish taxa (312 identified at species level, 49 identified at genus, subfamily and family) from 178 genera, 71 families and 2 classes (Teleostei and Elasmobranchii). The five benthic datasets comprise the description of the detection and cover of 81 taxa, 82 genera, 68 families, 15 classes, and 4 kingdoms belonging to Animalia, Bacteria, Plantae, Chromista. Because photographic identification does not always

allow the species level identification, the number of genera is similar to that of species richness, and the number of families is also comparatively high. In general, datasets with a large spatial coverage (dataset XIV from Sisbiota-Mar project) showed higher richness across taxa in comparison with local monitoring (dataset XVI) (Fig. 6).

Bare substrate, sediment, lost information (shade, quadrat, tape), morpho-anatomical benthic groups and turf were not included in the benthic datasets because they do not represent taxonomic entities in which DCS standards are based. For

datasets with benthic cover categories not included here (i.e. the raw data), please contact the data providers (see **Author contributions**).

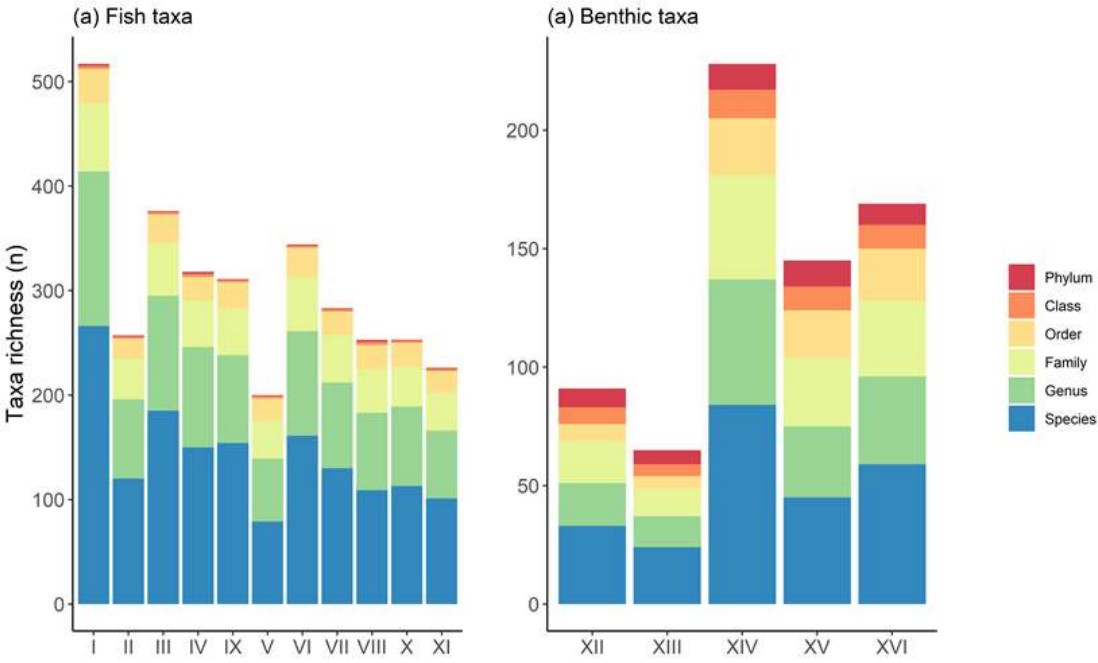

**Figure 6: Taxa richness according to taxonomic classification level foreach dataset. Column labels denote different reef fish and benthos datasets. Datasets: I - Fish communities from the Brazilian province, II - Abrolhos Bank**
**monitoring, III - Arraial do Cabo (Rio de Janeiro) monitoring, IV - Oceanic islands' monitoring, V - Fish**





**assemblages from Trindade and Martin Vaz, VI - Santa Catarina monitoring, VII - Fish assemblages from Guarapari, Espírito Santo, VIII - Fish assemblages from Southern Espírito Santo, IX - Trophic interactions along the Western Atlantic, X - Alcatrazes monitoring, XI - Rio Grande do Norte monitoring, XII - Benthic communities' monitoring in Abrolhos Bank, XIII - Extended benthic communities from the Brazilian province, XIV - Benthic**
**communities from the Brazilian province, XV - Benthic communities' monitoring from oceanic islands, XVI - Benthic communities from Rio Grande do Norte.**

### 2.8 Taxonomic coverage per group

#### 2.8.1 Fish

*Taxonomic ranks*

Kingdom: Animalia

Phylum: Chordata

Class: Elasmobranchii, Teleostei

Orders: Acanthuriformes, Acropomatiformes, Albuliformes, Anguilliformes, Aulopiformes, Batrachoidiformes, Beloniformes, Blenniiformes, Callionymiformes, Carangaria *incertae sedis*, Carangiformes, Carcharhiniformes,
Centrarchiformes, Clupeiformes, Dactylopteriformes, Elopiformes, Eupercaria *incertae sedis*, Gobiesociformes, Gobiiformes, Holocentriformes, Kurtiformes, Lophiiformes, Mugiliformes, Mulliformes, Myliobatiformes, Orectolobiformes, Ovalentaria *incertae sedis*, Perciformes, Pleuronectiformes, Rhinopristiformes, Scombriformes, Syngnathiformes, Tetraodontiformes, Torpediniformes

Families: Acanthuridae, Albulidae, Antennariidae, Apogonidae, Aulostomidae, Balistidae, Batrachoididae, Belonidae,
Blenniidae, Bothidae, Callionymidae, Carangidae, Carcharhinidae, Centropomidae, Chaenopsidae, Chaetodontidae, Cirrhitidae, Clupeidae, Dactylopteridae, Dasyatidae, Diodontidae, Dorosomatidae, Echeneidae, Elopidae, Engraulidae, Ephippidae, Fistulariidae, Gerreidae, Ginglymostomatidae, Gobiesocidae, Gobiidae, Grammatidae, Gymnuridae, Haemulidae, Hemiramphidae, Holocentridae, Kyphosidae, Labridae, Labrisomidae, Lutjanidae, Malacanthidae, Microdesmidae, Monacanthidae, Mugilidae, Mullidae, Muraenidae, Myliobatidae, Narcinidae, Ogcocephalidae,
Ophichthidae, Opistognathidae, Ostraciidae, Paralichthyidae, Pempheridae, Pinguipedidae, Pomacanthidae, Pomacentridae, Pomatomidae, Priacanthidae, Rachycentridae, Rhinobatidae, Sciaenidae, Scombridae, Scorpaenidae, Serranidae, Sparidae, Sphyraenidae, Syngnathidae, Synodontidae, Tetraodontidae, Tripterygiidae

Genus: *Ablennes, Abudefduf, Acanthistius, Acanthostracion, Acanthurus, Aetobatus, Ahlia, Albula, Alectis, Alphestes, Aluterus, Amblycirrhitus, Amphichthys, Anchoviella, Anisotremus, Antennarius, Apogon, Archosargus, Arcos, Astrapogon,*
*Aulostomus, Azurina, Balistes, Bathygobius, Bathytoshia, Belone, Bodianus, Bothus, Brachygenys, Calamus, Callionymus,*



*Cantherhines, Canthidermis, Canthigaster, Caranx, Carcharhinus, Centropomus, Centropyge, Cephalopholis, Chaetodipterus, Chaetodon, Chilomycterus, Chloroscombrus, Choranthias, Chromis, Clepticus, Coryphopterus, Cosmocampus, Cryptotomus, Ctenogobius, Dactylopterus, Dasyatis, Decapterus, Dermatolepis, Diapterus, Diodon, Diplectrum, Diplodus, Doratonotus, Dules, Echeneis, Echidna, Elacatinus, Elagatis, Elops, Emblemariopsis, Enchelycore,*

*Engraulis, Enneanectes, Entomacrodus, Epinephelus, Equetus, Eucinostomus, Eugerres, Euthynnus, Fistularia, Gerres, Ginglymostoma, Gnatholepis, Gobioclinus, Gobiosoma, Goblioclinus, Gramma, Gymnothorax, Gymnura, Haemulon, Halichoeres, Harengula, Hemiramphus, Heteropriacanthus, Hippocampus, Holacanthus, Holocentrus, Hypanus, Hypleurochilus, Hyporthodus, Hypsoblennius, Kyphosus, Labrisomus, Lactophrys, Lutjanus, Malacanthus, Malacoctenus, Melichthys, Micrognathus, Microgobius, Microspathodon, Monacanthus, Mugil, Mulloidichthys, Mullus, Muraena,*

*Mycteroperca, Myliobatis, Myrichthys, Myripristis, Narcine, Negaprion, Nicholsina, Ocyurus, Odontoscion, Ogcocephalus, Ophichthus, Ophioblennius, Opistognathus, Orthopristis, Pagrus, Parablennius, Paraclinus, Paradiplogrammus, Paralichthys, Paranthias, Pareques, Pempheris, Phaeoptyx, Pinguipes, Platybelone, Plectrypops, Pomacanthus, Pomatomus, Porichthys, Priacanthus, Prognathodes, Pronotogrammus, Pseudobatos, Pseudocaranx, Pseudupeneus, Ptereleotris, Rachycentron, Rhinobatos, Rhomboplites, Rypticus, Sardinella, Sargocentron, Scartella, Scarus,*

*Scomberomorus, Scorpaena, Scorpaenodes, Selar, Selene, Seriola, Serranus, Sparisoma, Sphoeroides, Sphyraena, Starksia, Stegastes, Stephanolepis, Strongylura, Synodus, Thalassoma, Trachinotus, Upeneus, Uraspis, Xyrichtys, Zapteryx*

Species: *Ablennes hians, Abudefduf saxatilis, Acanthistius brasilianus, Acanthostracion polygonium, Acanthostracion quadricornis, Acanthurus bahianus, Acanthurus chirurgus, Acanthurus coeruleus, Acanthurus monroviae, Aetobatus narinari, Ahlia egmontis, Albula vulpes, Alectis ciliaris, Alphestes afer, Aluterus monoceros, Aluterus scriptus,*

*Amblycirrhitus pinos, Amphichthys cryptocentrus, Anchoviella lepidentostole, Anisotremus moricandi, Anisotremus surinamensis, Anisotremus virginicus, Antennarius multiocellatus, Apogon americanus, Apogon pseudomaculatus, Archosargus probatocephalus, Archosargus rhomboidalis, Arcos rhodospilus, Astrapogon puncticulatus, Aulostomus maculatus, Aulostomus strigosus, Azurina multilineata, Balistes capriscus, Balistes vetula, Bathygobius soporator, Bathytoshia centroura, Belone belone, Bodianus insularis, Bodianus pulchellus, Bodianus rufus, Bothus lunatus, Bothus*

*ocellatus, Brachygenys chrysargyreum, Calamus calamus, Calamus penna, Callionymus bairdi, Cantherhines macrocerus, Cantherhines pullus, Canthidermis maculata, Canthidermis sufflamen, Canthigaster figueiredoi, Caranx bartholomaei, Caranx crysos, Caranx hippos, Caranx latus, Caranx lugubris, Caranx ruber, Carcharhinus perezii, Centropomus parallelus, Centropomus undecimalis, Centropyge aurantonotus, Cephalopholis fulva, Chaetodipterus faber, Chaetodon ocellatus, Chaetodon sedentarius, Chaetodon striatus, Chilomycterus reticulatus, Chilomycterus spinosus, Chloroscombrus*

*chrysurus, Choranthias salmopunctatus, Chromis flavicauda, Chromis jubauna, Chromis limbata, Chromis scotti, Clepticus brasiliensis, Coryphopterus dicrus, Coryphopterus glaucofraenum, Coryphopterus thrix, Cosmocampus albirostris, Cryptotomus roseus, Ctenogobius saepepallens, Dactylopterus volitans, Decapterus macarellus, Decapterus punctatus, Dermatolepis inermis, Diapterus auratus, Diodon holocanthus, Diodon hystrix, Diplectrum formosum, Diplectrum radiale,*





*Diplodus argenteus, Doratonotus megalepis, Dules auriga, Echeneis naucrates, Echidna catenata, Elacatinus figaro,*
*Elacatinus phthirophagus, Elacatinus pridisi, Elagatis bipinnulata, Elops saurus, Emblemariopsis signifer, Enchelycore*
*anatina, Enchelycore carychroa, Enchelycore nigricans, Engraulis anchoita, Enneanectes altivelis, Enneanectes smithi,*
*Epinephelus adscensionis, Epinephelus itajara, Epinephelus marginatus, Epinephelus morio, Equetus lanceolatus,*
*Eucinostomus argenteus, Eucinostomus lefroyi, Eucinostomus melanopterus, Eugerres brasilianus, Euthynnus alletteratus,*
*Fistularia petimba, Fistularia tabacaria, Gerres cinereus, Ginglymostoma cirratum, Gnatholepis thompsoni, Gobioclinus*
*kalisherae, Gobiosoma hemigymnum, Goblioclinus kalisherae, Gramma brasiliensis, Gymnothorax funebris, Gymnothorax*
*miliaris, Gymnothorax mordax, Gymnothorax moringa, Gymnothorax vicinus, Gymnura altavela, Haemulon atlanticus,*
*Haemulon aurolineatum, Haemulon chrysargyreum, Haemulon melanurum, Haemulon parra, Haemulon plumierii,*
*Haemulon squamipinna, Halichoeres bivittatus, Halichoeres brasiliensis, Halichoeres dimidiatus, Halichoeres maculipinna,*
*Halichoeres penrosei, Halichoeres poeyi, Halichoeres radiatus, Halichoeres rubrovirens, Halichoeres sazimai, Harengula*
*clupeola, Harengula jaguana, Hemiramphus brasiliensis, Heteropriacanthus cruentatus, Hippocampus reidi, Holacanthus*
*ciliaris, Holacanthus tricolor, Holocentrus adscensionis, Hypanus berthalutzae, Hypanus marianae, Hypleurochilus brasil,*
*Hypleurochilus fissicornis, Hypleurochilus pseudoaequipinnis, Hyporthodus niveatus, Hypsoblennius invemar, Kyphosus*
*bigibbus, Kyphosus cinerascens, Kyphosus sectatrix, Kyphosus vaigiensis, Labrisomus cricota, Labrisomus nuchipinnis,*
*Lactophrys trigonus, Lutjanus alexandrei, Lutjanus analis, Lutjanus apodus, Lutjanus cyanopterus, Lutjanus griseus,*
*Lutjanus jocu, Lutjanus synagris, Malacanthus plumieri, Malacoctenus brunoi, Malacoctenus delalandii, Malacoctenus*
*lianae, Malacoctenus triangulatus, Malacoctenus zaluari, Melichthys niger, Micrognathus crinitus, Microgobius carri,*
*Microspathodon chrysurus, Monacanthus ciliatus, Mugil curema, Mulloidichthys martinicus, Mullus argentinae, Muraena*
*melanotis, Muraena pavonina, Muraena retifera, Mycteroperca acutirostris, Mycteroperca bonaci, Mycteroperca*
*interstitialis, Mycteroperca microlepis, Mycteroperca venenosa, Myliobatis goodei, Myrichthys breviceps, Myrichthys*
*ocellatus, Myripristis jacobus, Narcine brasiliensis, Negaprion brevirostris, Nicholsina usta, Ocyurus chrysurus,*
*Odontoscion dentex, Ogcocephalus vespertilio, Ophichthus ophis, Ophioblennius atlanticus, Ophioblennius macclurei,*
*Ophioblennius trinitatis, Opistognathus aurifrons, Opistognathus whitehursti, Orthopristis ruber, Pagrus pagrus,*
*Parablennius marmoreus, Parablennius pilicornis, Paraclinus spectator, Paradiplogrammus bairdi, Paralichthys*
*brasiliensis, Paranthias furcifer, Pareques acuminatus, Pempheris schomburgkii, Phaeoptyx pigmentaria, Pinguipes*
*brasilianus, Platybelone argalus, Plectrypops retrospinis, Pomacanthus arcuatus, Pomacanthus paru, Pomatomus saltatrix,*
*Porichthys porosissimus, Priacanthus arenatus, Prognathodes brasiliensis, Prognathodes guyanensis, Prognathodes*
*obliquus, Pronotogrammus martinicensis, Pseudobatos horkelii, Pseudocaranx dentex, Pseudupeneus maculatus,*
*Ptereleotris randalli, Rachycentron canadum, Rhomboplites aurorubens, Rypticus bistrispinus, Rypticus saponaceus,*
*Sardinella brasiliensis, Sargocentron bullisi, Scartella cristata, Scarus trispinosus, Scarus zelindae, Scomberomorus*
*brasiliensis, Scomberomorus maculatus, Scomberomorus regalis, Scorpaena brachyptera, Scorpaena brasiliensis,*
*Scorpaena dispar, Scorpaena isthmensis, Scorpaena plumieri, Scorpaenodes caribbaeus, Selar crumenophthalmus, Selene*
*setapinnis, Selene vomer, Seriola dumerili, Seriola lalandi, Seriola rivoliana, Serranus aliceae, Serranus atricauda,*



*Serranus atrobranchus, Serranus baldwini, Serranus flaviventris, Serranus phoebe, Sparisoma amplum, Sparisoma axillare,*
*Sparisoma frondosum, Sparisoma radians, Sparisoma rocha, Sparisoma tuiupiranga, Sparisoma viride, Sphoeroides camila,*
*Sphoeroides greeleyi, Sphoeroides testudineus, Sphyraena barracuda, Sphyraena borealis, Sphyraena guachancho,*
*Sphyraena picudilla, Sphyraena tome, Starksia brasiliensis, Stegastes fuscus, Stegastes pictus, Stegastes rocasensis,*
*Stegastes sanctipauli, Stegastes trindadensis, Stegastes variabilis, Stephanolepis hispidus, Strongylura marina, Strongylura*
*timucu, Synodus foetens, Synodus intermedius, Synodus synodus, Thalassoma noronhanum, Trachinotus falcatus,*
*Trachinotus goodei, Trachinotus ovatus, Upeneus parvus, Uraspis secunda, Xyrichtys novacula, Zapteryx brevirostris*

Hybrid: *Menephorus, Menephorus dubius*, hybrid between *Cephalopholis fulva* and *Paranthias furcifer*.

### 2.8.2 Benthos

*Taxonomic ranks*

Kingdom: Animalia, Bacteria, Chromista, Plantae

Phylum: Annelida, Arthropoda, Bryozoa, Chlorophyta, Chordata, Cnidaria, Cyanobacteria, Echinodermata, Mollusca, Ochrophyta, Porifera, Rhodophyta

Class: Anthozoa, Ascidiacea, Asteroidea, Bivalvia, Demospongiae, Echinoidea, Florideophyceae, Gymnolaemata, Homoscleromorpha, Hydrozoa, Ophiuroidea, Phaeophyceae, Polychaeta, Thecostraca, Ulvophyceae

Orders: Actiniaria, Agelasida, Amphilepidida, Anthoathecata, Aplousobranchia, Bryopsidales, Camarodonta, Ceramiales,
Cheilostomatida, Chondrillida, Cladophorales, Clionaida, Corallinales, Dasycladales, Diadematoida, Dictyoceratida, Dictyotales, Ectocarpales, Fucales, Gelidiales, Gigartinales, Halymeniales, Haplosclerida, Homosclerophorida, Leptothecata, Malacalcyonacea, Nemaliales, Peyssonneliales, Phlebobranchia, Poecilosclerida, Rhodymeniales, Sabellida, Scleractinia, Stolidobranchia, Suberitida, Ulvales, Verongiida, Zoantharia

Families: Agariciidae, Agelasidae, Aglaopheniidae, Aplysinidae, Ascidiidae, Astrangiidae, Astrocoeniidae, Bryopsidaceae,
Callyspongiidae, Carijoidae, Caulerpaceae, Champiaceae, Chondrillidae, Cladophoraceae, Clionaidae, Codiaceae, Corallinaceae, Crambeidae, Cystocloniaceae, Dasycladaceae, Dendrophylliidae, Diadematidae, Dictyotaceae, Didemnidae, Echinometridae, Faviidae, Galaxauraceae, Gelidiaceae, Gelidiellaceae, Gorgoniidae, Halimedaceae, Halymeniaceae, Irciniidae, Lithophyllaceae, Lomentariaceae, Meandrinidae, Mesophyllumaceae, Microcionidae, Milleporidae, Montastraeidae, Nephtheidae, Niphatidae, Ophiotrichidae, Paramuriceidae, Parazoanthidae, Petrosiidae, Peyssonneliaceae,
Plakinidae, Plexaurellidae, Plexauridae, Pocilloporidae, Poritidae, Pterogorgiidae, Rhizangiidae, Rhodomelaceae,





Sargassaceae, Schizoporellidae, Scytosiphonaceae, Serpulidae, Sertulariidae, Siphonocladaceae, Sphenopidae, Styelidae, Suberitidae, Ulvaceae, Valoniaceae, Wrangeliaceae, Zoanthidae

Genus: *Agaricia, Agelas, Aiolochroia, Amphimedon, Amphiroa, Aplysina, Astrangia, Botrylloides, Bryopsis, Callyspongia,*
*Canistrocarpus, Carijoa, Caulerpa, Chaetomorpha, Champia, Chondrilla, Cladophora, Clathria, Cliona, Codium, Colpomenia, Diadema, Dictyopteris, Dictyosphaeria, Dictyota, Didemnum, Digenea, Echinometra, Favia, Galaxaura, Gelidiella, Gelidiopsis, Gelidium, Halimeda, Halymenia, Heterogorgia, Hypnea, Idiellana, Ircinia, Jania, Laurencia, Leptogorgia, Lobophora, Macrorhynchia, Madracis, Meandrina, Mesophyllum, Millepora, Monanchora, Montastraea, Muricea, Muriceopsis, Mussismilia, Neomeris, Neospongodes, Ophiothela, Padina, Palythoa, Parazoanthus, Peyssonnelia, Phallusia, Phyllogorgia, Plakinastrella, Plexaurella, Porites, Protopalythoa, Pseudosuberites, Sargassum, Schizoporella,*
*Siderastrea, Stephanocoenia, Stypopodium, Tricleocarpa, Trididemnum, Tubastraea, Udotea, Ulva, Valonia, Verongula, Wrangelia, Xestospongia, Zoanthus*

Species: *Agaricia agaricites, Agaricia fragilis, Agaricia humilis, Agelas dispar, Aiolochroia crassa, Aplysina fulva, Aplysina lactuca, Aplysina lacunosa, Astrangia rathbuni, Astrangia solitaria, Botrylloides nigrum, Bryopsis pennata, Callyspongia vaginalis, Canistrocarpus cervicornis, Carijoa riisei, Caulerpa cupressoides, Caulerpa mexicana, Caulerpa racemosa,*
*Caulerpa verticillata, Champia parvula, Chondrilla nucula, Cliona delitrix, Codium intertextum, Colpomenia sinuosa, Diadema antillarum, Dictyopteris jolyana, Dictyopteris justii, Dictyopteris plagiogramma, Dictyosphaeria versluysii, Dictyota menstrualis, Dictyota mertensii, Didemnum perlucidum, Digenea simplex, Echinometra lucunter, Favia gravida, Gelidiella acerosa, Gelidium floridanum, Gelidium pusillum, Halimeda discoidea, Halimeda opuntia, Hypnea musciformis, Idiellana pristis, Ircinia felix, Ircinia strobilina, Lobophora variegata, Macrorhynchia philippina, Madracis decactis,*
*Meandrina brasiliensis, Mesophyllum erubescens, Millepora alcicornis, Millepora braziliensis, Millepora nitida, Monanchora arbuscula, Monanchora brasiliensis, Montastraea cavernosa, Muricea flamma, Muriceopsis sulphurea, Mussismilia braziliensis, Mussismilia harttii, Mussismilia hispida, Mussismilia leptophylla, Neomeris annulata, Neospongodes atlantica, Ophiothela mirabilis, Palythoa caribaeorum, Palythoa grandiflora, Palythoa variabilis, Phallusia nigra, Phyllogorgia dilatata, Plakinastrella microspiculifera, Plexaurella grandiflora, Plexaurella regia, Porites astreoides,*
*Porites branneri, Stephanocoenia intersepta, Tricleocarpa cylindrica, Valonia ventricosa, Verongula gigantea, Verongula rigida, Xestospongia muta, Zoanthus sociatus.*

## 3 Potential use and conclusions

Here, we presented standardized datasets for reef fish and benthos of the Brazilian marine biogeographical province. These datasets were predominantly used in separate analyses and publications (Aued et al., 2018; Francini-Filho et al., 2013; Longo
et al., 2019; Morais et al., 2017; Santana et al., 2023), and in some cases were already published in online data platforms with varied formats (Aued et al., 2018; Longo et al., 2019; Santana et al., 2023). Standardizing different datasets under the





same settings enables them to be the basis for analysis of spatial and temporal trends at population and community scales. The aggregation of these datasets comprises data collected across more than 50 locations and 350 localities, and some monitored for over 20 years. Thus, these datasets provide society, researchers and other interested parties (managers) the

result of two decades of federal and state funding research focusing on Brazilian reefs. Unified, organized and under full public access, these datasets can be used by independent researchers working with diverse questions and analytical perspectives regarding reef ecology and conservation. These datasets will hopefully foster the inclusion of Brazilian reefs in future global synthesis studies, as many previous studies did not use data available for Brazilian reefs (Pellowe et al., 2023; Strona et al., 2021), or included only a few locations (Tebbett et al., 2023). A number of analyses using these datasets have

been already published by our group (e.g. (Francini-Filho and De Moura, 2008), using dataset number II; (Francini-Filho et al., 2013), using dataset number XII; (Luza et al., 2022), using datasets IX and XIV; (Luza et al., 2023b), using datasets I and XIV; (Santana et al., 2023), using datasets XIII and XIV; (Waechter et al., 2024), using dataset I and a new dataset of fish aesthetic value) and others are in progress (Dambros et al. unpublished data using all the 16 datasets, and Luza et al. unpublished data, using the 11 fish data sets). The aggregation or integration of different fish datasets (see for instance the

approaches using multiple datasets described in (Fletcher et al., 2019)) can contribute to a broader picture of fish biomass distribution across the coast and oceanic islands and also over time (Morais et al., 2017). The distribution of fish biomass can be analysed in terms of locality position relative to Marine Protected Areas, Conservation Priority Areas, and fishing areas (Graham et al., 2017). Nonetheless, the distribution and cover/abundance of benthic organisms relative to these factors might show the overall health of Brazilian reefs along with variation in major oceanographic conditions (e.g. temperature, marine

heatwaves) to answer questions about climate change (Vergés et al., 2014). The proper use of these datasets will enable the formulation of management recommendations for public policies focusing on reef protection, conservation and sustainable use at local and regional levels.

## 4 Data availability

These data are published under CC-BY 4.0 license.

**Table 3. Data set DOIs.**

| Group / Data set | DOI | Citation |
|---|---|---|
| Reef fish | | |
| Dataset I: Fish communities from the Brazilian province | https://doi.org/10.25607/7nxv5v | (Morais et al., 2024) |
| Dataset II: Abrolhos Bank monitoring | https://doi.org/10.25607/jqwg40 | (Francini-Filho, 2024a) |
| Dataset III: Arraial do Cabo (Rio de Janeiro) monitoring* | https://doi.org/10.25607/xdh1iv | (Ferreira et al., 2024) |
| Dataset IV: Oceanic islands' monitoring | http://doi.org/10.25607/rov4or | (Cordeiro et al., 2021) |
| Dataset V: Fish assemblages from Trindade and Martin Vaz | https://doi.org/10.25607/vvjwcv | (Pinheiro, 2024) |
| Dataset VI: Santa Catarina monitoring | https://doi.org/10.25607/ys9koa | (Quimbayo et al., 2024) |





| | | |
|---|---|---|
| Dataset VII: Fish assemblages from Guarapari, Espírito Santo | https://doi.org/10.25607/qyfwlo | (Pinheiro and Simon, 2024a) |
| Dataset VIII: Fish assemblages from Southern Espírito Santo | https://doi.org/10.25607/e8dont | (Pinheiro and Simon, 2024b) |
| Dataset IX: Trophic interactions along the Western Atlantic | https://doi.org/10.25607/7vomv6 | (Longo and Inagaki, 2024) |
| Dataset X: Alcatrazes monitoring | https://doi.org/10.25607/4e5fuo | (Mendes et al., 2024) |
| Dataset XI: Rio Grande do Norte monitoring | https://doi.org/10.25607/2doybv | (Longo et al., 2024) |
| Benthos | | |
| Dataset XII: Benthic communities' monitoring in Abrolhos Bank | https://doi.org/10.25607/jjm2eh | (Francini-Filho, 2024b) |
| Dataset XIII: Extended benthic communities from the Brazilian province | https://doi.org/10.25607/tbymqb | (Santana et al., 2024) |
| Dataset XIV: Benthic communities from the Brazilian province | https://doi.org/10.25607/yr6dkc | (Aued et al., 2024) |
| Dataset XV: Benthic communities' monitoring from oceanic islands | http://doi.org/10.25607/yrfths | (Cordeiro et al., 2022) |
| Dataset XVI: Benthic communities from Rio Grande do Norte | https://doi.org/10.25607/c2c37e | (Longo and Roos, 2024) |

\* This dataset will be available on 05/01/2025.

**5 Code availability**

R codes used to format the data sets are available at: https://github.com/Sinbiose-Reefs/ReefSYN_data.git

**6 Author contributions**

A.L. Luza, C.A.M.M. Cordeiro standardized the data to the Darwin Core Standard and together with M.G. Bender wrote the first manuscript draft. All other authors shared datasets, acquired resources for conducting fieldwork, and had a critical contribution during data curation, formatting, and manuscript writing. Data owners: C.A.M.M. Cordeiro, A.W. Aued, B. Segal, C.E.L. Ferreira, T.C. Mendes, N.C. Roos, G.O. Longo, R.B. Francini-Filho, S.R. Floeter, H.T. Pinheiro, J-C. Joyeux, J.P. Quimbayo.

**7 Competing interests**

The authors declare that they have no conflict of interest.

**8 Disclaimer**

Embargoed dataset: III: Time series of Arraial do Cabo, Rio de Janeiro (DOI available on 05 January 2025).



## 9 Permits

Data were collected following Brazilian government legislation. This includes authorization to the Sisbiota-Mar project to assess images of the benthic communities in protected reefs, under the permits # 06/2012 (Parcel do Manuel Luis; SEMA-MA), # 29953–1 (Rocas Atoll; ICMBio/ MMA—Brazilian Ministry of Environment), # 29687–2 (Fernando de Noronha; ICMBio/ MMA—Brazilian Ministry of Environment), # 32145–1 (Costa dos Corais, ICMBio/ MMA—Brazilian Ministry of Environment), # 22637 (Abrolhos, ICMBio/ MMA—Brazilian Ministry of Environment), # 4416–1 (Trindade Island,

ICMBio/ MMA—Brazilian Ministry of Environment), # 37869 (Alcatrazes, ICMBio/ MMA—Brazilian Ministry of Environment), # 21422 (Florianópolis Norte, ICMBio/ MMA—Brazilian Ministry of Environment), and for RN Maracajaú (APA dos Recifes de Corais, IDEMA-RN).

## 10 Acknowledgements

This research was conducted by the team and collaborators of the Reef Synthesis Working Group (ReefSYN) funded by the
Synthesis Center on Biodiversity and Ecosystem Services (SinBiose, CNPq, #442417/2019-5 to MGB). Researchers from the 'Brazilian Marine Biodiversity Research Network—SISBIOTA-Mar' (CNPq #563276/2010-0 and FAPESC #6308/2011-8 to SRF), 'Programa de Monitoramento de Longa Duração das Comunidades Recifais de Ilhas Oceânicas—PELD ILOC' (CNPq #441327/2020-6, to CELF) and Universidade Federal do Espírito Santo (Fundação de Amparo à Pesquisa do Espírito Santo, FAPES grant #38854660/2007) collected and shared datasets used in this research. The project of dataset VIII was
funded by Fundação Grupo O Boticário, Fundação SOS Mata Atlântica and ICMBio. ALL received post-doctoral fellowships from CNPq (#153024/2022-4, #164240/2021-7, #151228/2021-3, #152410/2020-1) and CAPES (PDPG-POSDOC, #88887.800011/2022-00). JPQ received post-doctoral fellowships from FAPESP (2018/21380-0 and 2021/09279-4). GOL is grateful to his research productivity scholarship provided by CNPq (#310517/2019-2 and 308072/2022-7), and to Serrapilheira Institute (Grant No. Serra-1708-15364) for continued research support. CELF, RBFF and SRF are grateful for
their research productivity scholarships provided by CNPq (#304004/2018-9 to CELF, #309651/2021-2 to RBFF, and #307340/2019-8 to SRF). HT Pinheiro acknowledges a CAPES scholarship for his master degree between 2008 and 2010. The group thanks Ana Paula Prates (Brazilian Ministry of the Environment) for kindly sharing the map of priority areas, and Leticia Costa-Lotufo and the ProspecMar-Ilhas team (CNPq, #62/2013) for supporting PELD researchers during data acquisition on oceanic islands.




## 11 Financial support

National Council for Scientific and Technological Development (Conselho Nacional de Desenvolvimento Científico e Tecnológico, CNPq).

Fundação de Amparo à Pesquisa do Estado do Espírito Santo (FAPES)

Fundação de Amparo à Pesquisa do Estado de Santa Catarina (FAPESC)

Fundação de Amparo à Pesquisa do Estado de São Paulo (FAPESP)

Fundação de Amparo à Pesquisa do Estado do Rio de Janeiro (FAPERJ)

Serrapilheira Institute

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
