# Peer review of "Standardized datasets of Brazilian reef diversity in space and time"

_Earth System Science Data, 2024_

## Community Comment (CC1)

**Review** of

"Standardized datasets of Brazilian reef diversity in space and time", submitted to
**Earth System Science Data**.

**Overall comments**:

Firstly, I would like to thank the journal and the authors for the opportunity to review this manuscript.

This is an excellent data paper, that will make relevant baseline data available for countless investigations on reef biodiversity and ecology in the Southwestern Atlantic. The authors make 16 standardized diversity datasets available, all of which are distributed across numerous sites in a broad latitudinal range in the Southwestern Atlantic. The manuscript is well-written, and it fits the journal scope very well.

The Introduction is generally well-structured, providing a thorough background on the transition in ecological research toward open data sharing and global collaboration. The Materials and Methods section reads well and needs only minor corrections. The Potential Use and Conclusions section need some improvement in highlighting the importance of these standardized datasets, and further depth on how they may aid in marine conservation, especially from a more applied perspective. However, this does not take away the relevance of the manuscript.

Overall, it is an excellent contribution, and I commend the authors for it. Therefore, I recommend publication after a minor revision.

Best regards

**Specific comments**:

Abstract

1. "The Brazilian marine biogeographical province (SW Atlantic) hosts coral and rocky reefs that cover ~27 degrees of latitude and are distributed along a relatively narrow continental shelf and four oceanic islands and archipelagos." – I wouldn't call it narrow. It is narrow in its central portion, but quite wide in the in its southern and (especially) northern portions.

Introduction

2. Please provide line numbers for all lines to facilitate reviewing.

3. "As a consequence, the field has experienced a hike in multidisciplinary research…" – I would add more recent sources to show that this trend is ongoing and not solely historical.

4. The paragraph about the DCS is a bit too abrupt and could be more smoothly integrated. Perhaps by showing the importance of data standardization earlier, linking it as a necessary development that supports the move to open-access repositories.

5. The graphical quality of Fig. 1 needs improvement. I would also improve the captions, making them a bit more self-explanatory; for instance, how were the keywords in (B) obtained?

6. While the local context of Brazil's SinBiose and ReefSYN initiatives is well described, the broader global context could be emphasized. For instance, it would be helpful to discuss how these initiatives align with or contribute to global research efforts on biodiversity.

7. Expanding on how SinBiose's focus on Brazilian ecosystems adds unique value to global ecological research could strengthen the Introduction. For example, highlighting the distinctiveness of Brazilian reefs and the insights they provide into tropical reef ecosystems under environmental stress.

Materials and Methods

8. "Methodology" or "Methods"?

9. I would clearly state that the Brazilian Province is unique – there are no other similar provinces in the world.

10. What are "coralligenous reefs" and how are they different from rocky reefs, regardless, I would be more cautious here – Brazil has four types of reef environments: true coral reefs (coralligenous?), algal reefs, sandstone reefs, and rocky reefs.

11. Were only data originating from transects used?

12. The PELD-ILOC abbreviation has not been introduced to the reader.

13. Fig.2 legends could also be improved. Graphic quality also.

14. "The remaining datasets are spatial snapshots (only one visit to a locality) through which data were collected on different events over many years"- this sentence reads odd.

15. Table legends could also be improved – for instance, in Table 1, it does not state that the datasets are for the Southwestern Atlantic Ocean.

16. I would add the timespan of each monitoring effort to Tables 1 and 2.

17. I would recommend the authors to also use the Coral Vivo dataset, which spans almost 10 years of surveying in reefs located in the Porto Seguro area. Maybe also some datasets from researchers affiliated to the ReBentos network.

18. "Data were checked by two data managers (A.L. Luza, C. Cordeiro) and questions were sent to data owners whenever necessary. The data owners are listed in the Author contributions' section" – this is very unorthodox writing style. I would remove it, as well as some other passages that seem a bit too "personal". E.g. "please contact the data providers (see Author contributions)." – this should be placed in a Data Availability Statement, not in the M&M.

19. Section 2.8 does not belong in the manuscript, but in a supplementary information file.

Potential use and conclusions

20. In general, I feel that the conclusions are not selling very well the relevance of this manuscript. I see conclusions only in lines 579-587, and in a shallow manner. This is critical baseline data that may have a profound impact on reef conservation in Brazil and even in other parts of the planet as well. This should be much more explored.

21. Brazilian reefs are currently undergoing severe degradation – overfishing, pollution, coral bleaching, poor recovery. This is something that must be stated, as it highlights the relevance of this manuscript from a timely and urgent perspective.